



# Dust observations with antenna measurements and its prospects for observations with Parker Solar Probe and Solar Orbiter

Ingrid Mann[1], Libor Nouzák[2], Jakub Vaverka[2], Tarjei Antonsen[1], Åshild Fredriksen[1], Karine Issautier[3], David Malaspina[4], Nicole Meyer-Vernet[3], Jiří Pavlů[2], Zoltan Sternovsky[4], Joan Stude[5], Shengyi Ye[6], Arnaud Zaslavsky[3]

[1]Department of Physics and Technology, UiT The Arctic University of Norway, 9037, Tromsø, Norway
[2]Department of Surface and Plasma Science, Charles University, Prague, 180 00, Czech Republic
[3]LESIA - Observatoire de Paris, Université PSL, CNRS, Sorbonne Université, Université de Paris, 5 place Jules Janssen, 92195 Meudon, France
[4]Laboratory for Atmospheric and Space Physics, University of Colorado, Boulder, CO 80303, USA
[5]Deutsches Zentrum für Luft- und Raumfahrt, Institut für Physik der Atmosphäre, Oberpfaffenhofen, Germany
[6]Department of Physics and Astronomy, University of Iowa, Iowa city, 52242-1479, Iowa, USA
*Correspondence to*: Ingrid Mann (ingrid.b.mann@uit.no)

**Abstract.** The electric and magnetic field instrument suite FIELDS on board the NASA Parker Solar Probe and the radio and plasma waves instrument RPWS on the ESA Solar Orbiter mission that explore the inner heliosphere are sensitive to signals generated by dust impacts. Dust impacts were observed using electric field antennas on spacecraft since the 1980s and the method was recently used with a number of space missions to derive dust fluxes. Here, we consider the details of dust impacts, subsequent development of the impact generated plasma and how it produces the measured signals. We describe empirical approaches to characterise the signals and compare to a qualitative discussion of laboratory simulations to predict signal shapes for spacecraft measurements in the inner solar system. While the amount of charge production from a dust impact will be higher near the sun than observed in the interplanetary medium before, the amplitude of pulses will be lower because of the different recovery behaviour that varies with the plasma environment. The photocurrent, that is expected to be higher near the Sun, is found to have moderate influence on the spacecraft potential.

## 1 Introduction

The space missions Parker Solar Probe and Solar Orbiter explore the inner heliosphere and close vicinity of the Sun and carry antennas experiments that respond to dust impacts onto the spacecraft. Parker Solar Probe (Parker Probe) is a NASA mission that was launched in August 2018 and will explore the vicinity of the Sun at closest distance ~ 10 solar radii near the solar equator. The mission payload includes the electric and magnetic fields instrument suite FIELDS (Bale et al. 2016). Solar Orbiter is an ESA mission with a launch planned in 2020. It



explores the vicinity of the Sun as close as 0.3 AU and at maximum 35° inclination from the Solar equatorial plane, and includes the RPW experiment (Maksimovic et al., 2019). Dust impacts were observed with electric antennas for field measurements since the 1980s with the Voyager mission (cf. Gurnett et al., 1997, Meyer-Vernet, 2001).

The method was recently used with a number of space missions to derive dust fluxes. While antenna measurements do not replace those of dedicated dust detectors, they are interesting because more space missions carry electric field instruments than carry dedicated dust detectors- In addition, antenna measurements detect lower dust fluxes because of their large collecting area of the whole spacecraft in comparison to the small collecting area of a dedicated dust detector. A limitation of the antenna measurements is that they do not provide information on dust

composition, only limited, if at all information on impact direction and dust mass. These derived values are highly uncertain. The relationship between dust impacts and the signals they produce in electric field instruments has also been considered in new instrument development and laboratory measurements.

Cosmic dust particles are one of the major constituents of the interplanetary medium in the inner heliosphere and

knowledge on dust near the Sun is so far based on remote observations and model assumptions. An exception are the measurements of the HELIOS mission with two spacecraft that reached a minimum distance 0.31 AU from the Sun and each carried a dust detector (Grün et al. 1980). Our basic understanding (see, e.g. Mann et al. 2004) is that large (> micrometre) dust particles that are fragments of comets and asteroids are in Keplerian orbits around the Sun. Their velocities and number densities increase with decreasing distance from the Sun. Fragments are produced

in dust-dust collisions for which the rates increase with decreasing distance from the Sun. The majority of fragments smaller than micrometre are pushed outward by radiation pressure and electro-magnetic forces. In addition to the interplanetary dust, interstellar dust particles stream into the inner heliosphere from interstellar upstream direction ~parallel to the ecliptic plane. Because of repulsion by the radiation pressure force, only the large interstellar dust reaches the inner heliosphere. A large fraction of dust is destroyed in the inner heliosphere, in sublimation and

other destruction processes and this generates a dust-free zone. Sun-gazing comets are a local highly variable source for dust particles (cf. Fig. 1). Dust material is released in the ambient solar wind, a process which is not well quantified. The solid fragments that are not fully destroyed are pushed outward and produce the small size portion of the interplanetary dust flux observed near Earth orbit. Parker Solar Probe and Solar Orbiter will for the first time explore the inner heliosphere in-situ. The dust impacts on the spacecraft will influence electric field measurements

on these spacecraft and create an opportunity to study the dust environments of the inner heliosphere.



This paper discusses dust detection with electric field measurements based on recent observations from several spacecraft, and discusses the prospects of studying dust impacts with the two space missions to the inner heliosphere. We start by describing the impact ionisation of dust particles (section 2), followed by a qualitative

discussion of the impact process in the vicinity of the spacecraft (section 3) and resulting antenna signal shapes (section 4). Section 5 discusses the signals and observations made recently with other space missions. In section 6 we describe our current state of knowledge on dust in the inner heliosphere and in section 7 we discuss implications for observations with Parker Solar Probe and Solar Orbiter.

## 2 Dust Impact Ionisation

Dust impacts on the spacecraft body generate clouds of ions and electrons through the process known as impact ionisation. These impact-generated clouds could be considered a plasma, they can also contain neutrals and solid fragments. The following discussion uses the term impact cloud to avoid confusion with the surrounding plasma. One impact ionisation model which has been shown to give a good agreement with laboratory experiments in relatively thin targets, for speeds between the supersonic limit and some tens of km/s (cf. Drapatz and Michel,

1974), can be summarised as follows: A dust grain impacting onto the spacecraft at supersonic speeds, i.e. faster than the order of a few kilometres per second, produces a shock compression wave in the solid which vaporises and ionises the dust as well as some material of the target where an impact crater forms. Charge production can occur via thermal ionisation because of the temperature of the vapour or via surface processes. The initial ionisation is followed by recombination and thermalisation. The amount of residual ionisation can be obtained from laboratory

measurements of the charge production. Experimental results are described for the charge production $Q$, as a function of the dust mass, $m$ and speed, $v$ according to $Q = \xi m^\alpha v^\beta$, where $Q$ is given in Coulombs, the $m$ in kilograms and $v$ in km/s. The exponents $\alpha$ and $\beta$ are dimensionless and determined empirically. The constant $\xi$ gives the proportionality and, as parameters $\alpha$ and $\beta$, it is dependent on both impactor and target composition.

Charge production is often determined empirically in dust accelerator facilities, in recent years at those of the University of Stuttgart (REF) and the University of Colorado (Shu et al., 2012) for the range of impact velocities shown in Fig (Fig. laboratory). The parameters to describe impact charge production derived from observations vary strongly for different impact materials; $\alpha$ has reported values between 0.7 and 1 while $\beta$ has been measured between 2.5 and 6.2 (see e.g. Dietzel et al. 1973, Auer 2001, Collette et al. 2014, and references therein). An often-

used relationship for dust impacts on spacecraft is $Q \approx 0.7mv^{3.5}$, which was reported for aluminium targets





(McBride and McDonnell, 1999). It should be noted that the exponents also change for low impact energies (speeds below ~1 km/s and sizes below ~10 nm). For low energy collisions, where fragments of significant sizes compared to the initial impactor survive, there may be surface effects such as capacitive contact charging (see e.g. John et al. 1980). The exponent also changes at high impact energies for speeds above ~50 kms[-1]and dust sizes above $\sim 1\ \mu m$

(Auer 2001; Göller and Grün, 1989). Collette et al. (2014) measured the charge impact yield as a function of impact velocity for common materials used on spacecraft, they point out the need for dedicated studies for > 50 km/s impacts. For very large impact energies, where the impactor gets completely vapourised, surface effects are negligible and the charge generation can be modelled through hydrodynamic theory (Hornung and Kissel, 1994). Moreover, there is a dependence of impact angle on the charge generation (Schulz and Sugita, 2006; Collette et al.

2014). Based on spectroscopic analysis of 4.7 – 5.6 kms[-1] impact flashes Sugita et al. (1998) find temperatures of about 4000 K to 5000 K for the impact vapour cloud. Subsequent observations yield temperatures of 0.9 to 3 eV for impact speeds varying from 10 to 40 km/s (Miyachi et al. 2008). Laboratory measurements find for the impact vapour ion temperatures of about 5 eV at 4 kms[-1] impact speed and > 10 eV at 20 kms[-1] (Collette et al. 2016).

## 3 The Impact Process

Formation of the dust impact signal involves the dust impact process, the interaction of the impact cloud with the surrounding plasma and finally the detection by electric field measurement. At the most basic level, dust impacts on the spacecraft body generate clouds of free electrons and positive or negative ions. These charged particles are attracted to, or repulsed from, the spacecraft surface according to its electric potential relative to the surrounding ambient plasma. Charged particles from the impact cloud can be re-collected by the spacecraft or escape to free

space and generate a transient deviation from the equilibrium spacecraft surface potential. The potential change can be positive or negative, e.g., escaping electrons generate a positive signal. Electrons are significantly faster than ions, thus the signal generated by escaping electrons appears before the signal generated by escaping ions for the case when the spacecraft potential is not too large. The spacecraft potential relaxes back to the equilibrium value via interaction with ambient plasma according to $\Phi_{SC} \sim e^{-t/\tau}$, where $\tau$ is a characteristic relaxation time

(Meyer-Vernet, 1985).

The different phases of the impact process for various spacecraft potentials (slightly positive, zero, and slightly negative) are illustrated in Fig. 3. In the first phase (T1), at which the spacecraft is assumed to be in equilibrium potential, the impact occurs, and an impact cloud is generated (green). Some of the cloud particles may be re-



collected. The second phase (T2) is characterised by electron escape and partial recollection depending on the target's potential, yielding a short signal precursor (blue). The third phase (T3) is characterised by the ion escape, decreasing the spacecraft potential (red). The final phase (T4) is the relaxation phase when the spacecraft potential returns to the equilibrium value (orange). Individual time steps are summarised in Table 1 and sketched in Fig. 3. The ratio between escaping electrons and ions in phases T2 and T3 depend on the spacecraft potential. For example,

more electrons than ions leave for a negatively charged spacecraft (left part of Fig. 3).

### 3.1 Impact cloud generation and expansion - T1

Charged particles at a small distance from the spacecraft body still influence its potential. The change in the spacecraft potential can thus not be observed directly after impact cloud formation but when charged particles are recollected or escape far enough and/or are sufficiently shielded by the ambient plasma or photoelectrons that their

influence on the spacecraft potential is reduced (Meyer-Vernet et al. 2017). The number of escaping particles depends on initial impactor energy and velocity after the initial cloud expansion. It is possible to assume that impact cloud electrons move in random directions due to collisions with ions. This implies that half of the electrons move towards the spacecraft before they are influenced by spacecraft potential, whereas the other half moving initially outwards is recollected if the target potential is positive and higher than their temperature (in eV). An alternative

model assumes approximately half of both the electrons and the ions move towards the spacecraft which corresponds to assuming thermal ionisation in the volume of initially neutral vapour. This assumption is in better agreement with the results of recent laboratory measurements (Nouzák et al., 2018, see below).

### 3.2 Electron escape - T2

The first part of the signal is generated by electrons escaping from the spacecraft body. The amplitude of the

electron signal is reduced when the spacecraft is charged positively because some electrons are attracted back to the spacecraft. All electrons are re-collected when the positive spacecraft potential is significantly higher than the temperature of electrons (no electron part in the signal). This is a very fast process (µs) and the characteristic time depends on a number of parameters. Independent from the ambient plasma this process is influenced by the geometry of the system and specifics of the antenna and parts of the spacecraft body as well as the energy (velocity)

of the electrons. The cloud expansion and internal shielding depends on the size of cloud formed by the impact. In space, the ambient plasma Debye length and magnitude of the photoemission current from the spacecraft determine the length scale of the spacecraft potential influence on the expansion.



### 3.3 Ion escape - T3

The ion escape works in a similar manner as electron escape, but happens at a lower rate if electron and ion
temperature are comparable. The potential induced by ions on the spacecraft and antennas is progressively shielded
by the ambient plasma electrons and photoelectrons. Moreover, escaping electrons can drag some of the ions behind
them. This can result in double population of escaping ions: fast and slow. All ions are recollected when the negative
spacecraft potential is significantly higher than the temperature of ions.

### 3.4 Relaxation - T4

The spacecraft potential returns back to the equilibrium value due to interaction with ambient plasma. The
relaxation time is determined by the ambient environment (plasma density, temperature, photoemission), and by
the capacitance of the spacecraft and antennas. On the other hand, higher plasma density and stronger
photoemission result in stronger currents from ambient plasma and thus in a significantly shorter relaxation time.
The relaxation time could be comparable or shorter than the ion escape or shielding time in dense plasma
environments (or under strong photoemission), and by the capacitance of the spacecraft and antennas. This will
result in a reduction of the detected signal and lowers the sensitivity of dust detection via electric field antennas.
Relaxation time can also be reduced by active experiments, for example by ASPOC (Active Spacecraft POtential
Control) (Vaverka et al., 2017b).

### 4 Antenna Signal Shapes

Electric field antennas can be operated as a dipole, where the voltage difference between two antenna booms is
measured, or a monopole, where the voltage difference between an antenna boom and the spacecraft body is
measured. It has been noted that the power spectral density of dust impact signals measured by monopole antenna
is significantly larger than that measured by the dipole antennas (Meyer-Vernet 1985, Tsintikidis et al., 1994;
Meyer-Vernet et al., 2014), and this difference is attributed to the low sensitivity of a symmetric dipole antenna to
dust impacts on spacecraft body. It is important to note that dust impact on a spacecraft body described by this
model, can be detected by the monopole electric field antenna as a potential drop between the spacecraft body and
one antenna. A dipole configuration measuring electric field as a potential difference between two antennas can be
utilized to detect a signal only when escaping electrons or ions influence the potential of one of the dipole antennas
asymmetrically. The described model shows a strong dependence on the spacecraft potential. This can be compared





with laboratory experiments for various polarity and sizes of bias voltage. A series of such measurement campaigns

have been performed at the dust accelerator facility at the University of Colorado in order to aid the interpretation

of signals collected in space. Collette et al. (2015) successfully identified different mechanisms of voltage signals

generation on the antennas. The experiments performed by Nouzák et al. (2018) have used a scale model of the

Cassini spacecraft to investigate the differences between antennas operated in monopole vs. dipole modes. The

results show that in the dipole mode the antennas are greatly insensitive to dust impacts occurring on the spacecraft

and only impacts on the antennas generate clear signals. This study helped clarifying the appropriate cross section

to be used for calculating the density of dust populations encountered by the spacecraft (Ye et al., 2016).

A few cases of impact events are shown in Fig. 4 derived from laboratory studies on scaled down Cassini model

(Nouzák et al., 2018). Although the signals are measured in dipole configuration, since the dust impacts one of the

dipole antennas, this configuration corresponds to monopole measurement when dust impacts the spacecraft body

as described above.

Figure 4 shows different signal shapes measured in the Cassini laboratory simulation and the signal development

of the different stages are described for each case in Table 2:

- The signal shown in panel (A) is for a strongly negatively biased target potential. All electrons are
  repulsed from the spacecraft and all ions are recollected back to a strongly negatively biased target. The
  ion escape part (red) is not apparent in this case. The electron part (blue) is followed directly by
  relaxation (orange).

- Panel (B) describes the signal shape for a reduced negative target potential. The number of escaping ions
  increases with the reduction of the negative potential (panel B). The electron part (blue) is followed by
  the smaller ion part (red) and relaxation (orange). A kink appears between the ion part (red) and
  relaxation (orange) see a left panel in Fig. 3.

- Panel (C) describing the signal measured at an unbiased target shows that similar numbers of electrons
and ions escape. The amplitude of the electron part (blue) is similar to the ion part (red).

- Panel (D) shows the case of positively charged target. The number of escaping electrons is reduced and
  the ion part of the signal exceeds the electron one. This results in a bipolar pulse where the first part is
  typically called "pre-spike" (Collette et al. 2015, Thayer et al. 2016). A larger number of escaping ions
  change the polarity of the pulse.



- Panel (DD) shows the signal for a higher positive target bias potential- The first (electron) part of the bipolar pulse is reduced with increasing positive target potential.

- Panel (E) shows a case of even higher positive bias. All electrons are re-collected in this case. The signal has no electron (blue) part and it has no "pre-spikes" in this case.

The shapes of all pulses measured in the laboratory for various biases can be explained by the model described above. It must be noted, however, that since electron and ion escapes are very fast processes (~µs), detection of a detailed structure of initial parts of pulses including "pre-spikes" thus requires fast electronics (sampling in the order of 100 kHz). Therefore, not all spacecraft are able to detect them, and a thorough inquiry into signal shapes using in-situ data is difficult.

## 5 Antenna signals observed in previous space missions

Detection of dust impacts with antenna measurements has recently been done in several space missions. In the following, we discuss the major findings related to dust detection from the respective missions.

### 5.1 STEREO

STEREO is a NASA mission that was launched on October 26, 2006, with the study of coronal mass ejections as primary science goal. The mission consists of two twin spacecraft that orbit the Sun at around 1 AU, one trailing the Earth (STEREO B) while the other leads (STEREO A). The study of the STEREO/WAVES radio receiver data proved to be of great interest for dust studies. STEREO/WAVES measured the flux of sub- micrometre dust near 1 AU (Meyer-Vernet et al. 2009, Belheouane et al. 2012, Zaslavsky et al. 2012) and discovered a highly time-variable flux of dust with size few nm (Meyer-Vernet et al., 2009). The nanodust impacts were observed frequently on both STEREO spacecraft as radio pulses on single monopole antennas. The physical mechanism that leads to their generation is not yet fully understood. The voltage was much higher on the antenna that was adequately located to be sensitive to impacts of prograde nanodust on each spacecraft (Meyer-Vernet et al. 2009), which destabilized the photoelectron sheath of that antenna (Pantellini et al., 2012), producing a ratio between antenna voltages in agreement with the mechanism producing the pulses (Zaslavsky et al. 2012). The formation of the signal involves a transient local perturbation of the photoelectron equilibrium current on the antenna being close to the impact. The steps that lead to the antenna signals have been studied with plasma simulations and semi-empirically (see e.g. Pantellini et al., 2012; Meyer-Vernet et al., 2014; Zaslavsky, 2015). Kellogg et al. 2018 suggested that STEREO does not observe nanodust, but did not propose an alternative mechanism able to explain the observations. The





larger dust impacts observed with STEREO/WAVES are observed with similar amplitudes at all three antennas. Based on STEREO/WAVES Zaslavsky (2015) proposed a model accounting for electric pulses generation by
electron collection after an impact, linking the shape and amplitude of the electric signals to the dust and local plasma parameters. Figure 5 shows the model applied to typical impact clouds.

**5.2 Cluster**

The Cluster mission launched in 2000 consists of four identical spacecraft orbiting the Earth in close formation. The highly elliptical orbit (4–20 Earth radii) crosses various parts of the Earth's magnetosphere. Each spacecraft is
equipped with two pairs of dipole electric field sensors (on 88 m booms tip-to-tip) (Gustafsson et al., 2001). Wide Band Data (WBD) instrument provide data of single electric or magnetic field component with a high sampling frequency in three modes (27.4 kHz, 54.9 kHz, and 219.5 kHz) (Gurnett et al., 1997). This resolution is sufficient to detect signals triggered by dust impacts. The dipole configuration is not sensitive to dust impacts on the spacecraft body. Some signal can be detected only after a direct dust impact on the one of the antennas or when the
expanding impact cloud influences the potential of the antenna. On the other hand, Cluster 1 operates with the only one remaining probe in the monopole configuration since 2009 (three probes have been lost during time). This situation makes the detection of dust impacts by the Cluster 1 spacecraft possible (Vaverka et al., 2017a). On the other hand, a presence of a large number of natural waves including electrostatic solitary waves in the Earth's magnetosphere significantly complicates such detection (Vaverka et al., 2018). The fact that solitary waves are
much more numerous than the expected amount of detected dust grains makes a reliable detection of dust impacts by the Cluster spacecraft very challenging. For this reason, the Cluster spacecraft are not optimal for dust studies.

**5.3 MMS**

The MMS mission consists of four Earth-orbiting spacecraft lunched in 2015 (Burch et al., 2016). While the missions are similar, the MMS electric field instruments just slightly differ from the Cluster ones. Each of the
spacecraft is equipped with three pairs of electric field probes, two in the spin plane (120 m tip-to-tip) and one in the axial plane (29 m, Torbert et al., 2016). The electric field is measured in the dipole configuration in all three directions with sampling frequency up to 8 kHz (burst mode) and up to 256 kHz in wave burst mode. The main difference is that the instrument operates simultaneously also in the monopole configuration. The combination of dipole and monopole measurements provides a complex information about the ambient electric field and the
spacecraft potential which is possible to use for the reliable identification of dust impacts. Solitary waves and other



structures in the ambient plasma or electric field generate simultaneously pulses both in monopole and dipole configuration. On the other hand, changes in the spacecraft potential triggered by the dust impact generate identical pulses on all monopole antennas and no signal in the dipole configuration (electric field data). This allows us reliably distinguish changes in the spacecraft potential from the other pulses as solitary waves (see Vaverka et al., 2018). A measurement with MMS, shown in Fig. 6, illustrates the different detections in monopole and dipole configuration.

### 5.4 Maven

MAVEN is a NASA mission to Mars. It launched on November 18, 2013 and arrived at Mars on September 22, 2014. MAVEN is designed to study the escape of Mars's atmosphere, including the contribution of plasma processes associated with the interaction between the solar wind and the planet (Jakosky et al. 2015). Voltage spikes consistent with the impact of micron dust on the spacecraft were detected by the MAVEN LPW (Langmuir Probe and Waves) experiment at orbital altitudes between 200 km and 1500 km (Andersson et al. 2015a). Andrews et al. 2015 found large variations in plasma density and spacecraft surface charging encountered by MAVEN as it dipped into the Martian ionosphere. This resulted in strong variation in the detectability of dust impact voltage spikes (Andersson et al. 2015b). Once these effects were taken into consideration, the estimated near-Mars micron dust flux observed by MAVEN was found to be consistent with the interplanetary dust flux expected at Mars. No evidence for moon-related dust rings or dust lofted from the surface (e.g. Sanchez-Lavega et al. 2015) was found with MAVEN LPW.

### 5.5 Wind

The NASA Wind spacecraft launched in November 1994 with the goal of studying the solar wind upstream of Earth. From 1994 to 2004, Wind executed a series of high apogee (100 $R_e$) orbits about Earth and several lunar flybys before being stationed in an orbit about the first Lagrange point ($L_1$) ~250 Re Sunward of Earth, where it remains operational to the present day (2019). The Wind WAVES experiment (Bougeret et al. 1995) detects voltage spikes consistent with the impact of micron-sized dust on the spacecraft body (Malaspina et al. 2014). These dust spikes are observable even though Wind WAVES makes only dipole electric field measurements, likely due to strong asymmetries of the dust impact signal on oppositely mounted antennas. Further, the rapid spin of the Wind spacecraft (one rotation every 3s) and asymmetry of dust impact voltage signals on the electric field wire antennas allows a crude directionality of the dust to be determined (Malaspina et al. 2014, Malaspina and Wilson 2016). The





observed amplitude and polarity of such signals are consistent with voltage induced on the antennas by positive

ions produced by impacts on the spacecraft, after it has recollected the electrons (Meyer-Vernet et al. 2014); this

new mechanism explained the previously unexplained voltage sign and amplitude for interstellar dust impacts on

Wind, and also the absence of nanodust detection on this spacecraft. The yearly modulation of Wind-observed

impacts was found to be consistent with the yearly variation in interplanetary micron dust (Malaspina et al. 2014,

Wood et al. 2015). Further supporting this conclusion was the observation that both Wind and STEREO observe

the same yearly modulation of interstellar dust flux (Kellogg et al. 2016). The long duration of the Wind mission

(> 25 years, over two full solar cycles) presents a unique opportunity to study how the solar magnetic field

modulates the entry of interstellar dust into the solar system and its arrival at 1 AU. To facilitate such studies, a

database cataloguing all dust impacts observed by Wind was created (Malaspina and Wilson 2016) and made

publicly available through the NASA Space Physics Data Facility Coordinated Data Analysis Web (CDAWeb)

(https://cdaweb.sci.gsfc.nasa.gov/index.html/).

**5.6 Cassini**

The Cassini Radio and Plasma Wave Science (RPWS) instrument measures oscillating electric fields over the

frequency range 1 Hz to 16 MHz and magnetic fields in the range 1 Hz to 12 kHz (Gurnett et al., 2004). The

instrument uses three nearly orthogonal electric field antennas ($E_u$, $E_v$, $E_w$, each 10 m long and 2.86 cm in diameter)

and three orthogonal magnetic search coil antennas. The $E_u$ and $E_v$ antennas are often used together as a dipole

antenna and $E_w$ and the spacecraft body as a monopole antenna (Gurnett, 1998), both sensitive to dust impacts. The

south-polar plume of Enceladus was one of the top discoveries made Cassini mission. During the Enceladus plume

crossing, besides dust impact signals, RPWS detected plasma oscillations induced by dust impacts, the frequencies

of which are equal to the local plasma frequencies (Ye et al. 2014a), which can be explained by a beam-plasma

instability induced by the impact-produced electrons when their speed exceeds the thermal speed of the ambient

plasma (Meyer-Vernet et al. 2017). Comparison of observations (Ye et al., 2014b), showed that the dust density

profile measured by RPWS is consistent with that measured by the dedicated dust detector on board.

Figure 9 compares the vertical dust density profiles measured by RPWS Wideband Receiver (WBR) and the Cassini

Dust Analyzer (CDA) High Rate Detector (HRD) during the ring plane crossing on DOY 361, 2016. HRD uses

polarized foils for dust detection and can measure high impact rates of particles bigger than a size threshold that

depends on the impact speed (Srama et al.,2004). Discontinuities in the RPWS dust density profile are due to gain





changes of WBR. The CDA data showed consistent peak densities around 0.04 m⁻³ (threshold ~ 0.8 micron) during

the Ring Grazing orbits, less than one order of magnitude higher than the RPWS dust density, which is within the uncertainty limit of the method (Ye et al., 2014). The density peak measured by RPWS (FWHM 600 to 1000 km) is wider than that by CDA (averaged profile shows a FWHM of 475 km). This difference is discussed in detail in Ye et al. (2018a). The E ring density structure based on RPWS measurements has been shown to be consistent as well with that revealed by optical observation (Ye et al., 2016a).


Cassini also allowed a comparison of measurements in dipole and monopole configuration. The difference is clearly seen on Fig. 7 which shows the electric power spectrum measured by the Cassini Radio and Plasma Wave Science (RPWS) HFR receiver simultaneously in dipole (top) and monopole (bottom) mode in Saturn's E-ring at the first close approach of Enceladus (Meyer-Vernet et al. 2014). During the subsequent mission, the Wideband receiver

(WBR) of the RPWS instrument was switched from monopole mode to dipole mode at a ring plane crossing, so that the responses of these two antenna modes to dust impacts were compared, assuming the dust density and size distribution did not change across the ring plane (Ye et al., 2016). Figure 8 shows an RPWS wave power spectrogram, which covers a one-hour period around a ring plane crossing on DOY 001, 2016. As the antenna mode switched from monopole to dipole at the ring plane at ~10:30, the spectral power decrease was accompanied

by a significant decrease in the negative impact rates (blue) and the polarity ratio jumping to ~1. The spectral power is proportional to the product of impact rate and average voltage jump size squared (Meyer-Vernet, 1985). So, the difference in spectral power at the antenna switch could be due to either lower impact rate or smaller average voltage pulse size, or both.

Ye et al. (2016b) compared the data collected with these two antenna setups and found that the wave power spectral density observed by the monopole antenna is ≈ 10 dB higher than that observed by the dipole antenna. This does not necessarily mean that the monopole antenna is more sensitive to individual dust impacts, because direct comparison of the waveforms observed by these two antennas showed that the sizes of the voltage jumps induced by dust impacts are comparable. Comparison of the impact rates showed that the monopole antenna detects ≈ 10

times more dust impacts than the dipole antenna. This difference in impact rates is roughly in line with the difference in the effective impact areas of the spacecraft body and the dipole electric antenna. Detailed analysis showed that the polarity ratio of the impacts detected by the dipole antenna changes with the projected area ratio of the dipole antenna elements ($E_u$ and $E_v$) as the spacecraft rotates, providing strong evidence that the dipole mode detects primarily impacts on the antenna booms.






Cassini cruise measurements between 1 and 5 AU also enabled us to study the rise time of the impact ionisation pulses as a function of dust mass and of heliocentric distance (Meyer-Vernet et al. 2017), a quantity of great importance for future missions since it determines the frequency range and voltage amplitude for dust detection.

## 6 Dust in the inner heliosphere

Many dust observations describe the dust flux near Earth orbit and it is also described with a semi-empirical model (see e.g. Grün et al. 1985). There are, however, few observational results on dust in the heliosphere inside 1 AU. Large dust particles ("micron dust" $m > 10^{-14}$ kg) are mainly influenced by gravity force and move in Keplerian orbits superimposed by a slow migration inward caused by the Poynting-Robertson effect. For dust with masses $10^{-19}$ kg $< m < 10^{-14}$ kg ("beta meteoroids") the radiation pressure force is comparable to the gravitation and if they

are released (e.g. by collisions of larger dust) they move outward in hyperbolic orbits. For even smaller dust with $m < 10^{-19}$ kg (nanodust) electromagnetic forces prevail. The particles of sizes of few nanometre are deflected in the solar wind similarly to the pick-up of heavy ions in the solar wind (Mann et al., 2007). Detailed trajectory calculations (Czechowski and Mann, 2010) show that nanodust can be trapped in orbits with perihelia very close to the Sun instead of being ejected. Trapping conditions depend on a number of different parameters so that the

nanodust flux outward can vary in time (Czechowski and Mann, 2010, 2012). The nanodust flux can vary also due to other effects, like the variation of the source, e.g. when the dust flux is enhanced by a single collision event in the inner heliosphere, or due to the influence of the solar magnetic field structure (see Juhasz and Horanyi, 2013). The ejected particles are accelerated outward (cf. Fig. 10) assuming that the dust is released from initially circular orbits. In reality, the situation is more complex, for a number of reasons: parent objects can be in elliptic orbits and

fragments are released at different locations along the orbit (cf. Fig. 11). While estimates are made for time-stationary conditions, current sheet crossings change the trajectories and in reality, the magnetic field is time-variable. Also coronal mass ejections change the conditions pushing outward large fractions of nanodust and with speed reaching 1000km/s (Czechowski and Kleimann 2017). The dust formation by mutual collisions depends on the dust material compositions (Ishimoto and Mann, 1999; Mann and Czechowski, 2005) and it is enhanced when

such high streams occur. Sun grazing comets (cf. Jones et al. 2018) are another source of time variable dust flux.

Cosmic dust particles of all sizes also hit the Earth, though their flux is highly uncertain. Estimates of cosmic dust fluxes near 1 AU and onto Earth range over several orders of magnitude and are based on a number of different





assumptions (cf. e.g. Nesvorný et al. 2011, Mann et al., 2011; Plane, 2012). Based on present knowledge, the mass
flux from sub-micrometre dust is smaller than that of the larger dust, but it is variable on time-scales of present
atmospheric observational data and its flux is not quantified. Measurements in the inner solar system can be helpful
for estimating the dust flux onto Earth: for one, it measures the dust in the inner heliosphere that is pushed outward
and will cross Earth orbit. Secondly, it will improve our understanding on how the interstellar dust flux varies with
helio-ecliptic latitude and time, e.g. with solar cycle. Most dust particles smaller than 1 micrometre are pushed
outward by radiation pressure and are in hyperbolic trajectories. Direct detections are rare and the Ulysses dust
detector pointed away from the inner solar system during most of its trajectory such that beta-meteoroids could be
observed only during certain parts of the spacecraft orbit (e.g. Wehry and Mann, 1999). The dust flux near Earth
orbit can be estimated from meteor observations, crater statistics and measurements with dust detectors. Based on
these sources, an empirical polynomial mass distribution was found (Grün et al., 1985; Ceplecha et al., 1998).
Observations by the STEREO spacecraft allowed extending this distribution to smaller masses (Zaslavsky et al.,
2012, Meyer-Vernet et al., 2009, Malaspina et al. 2015).

Photon-based observational studies are constrained by large contributions from dust near Earth to the brightness
(Mann et al., 2004). Nonetheless, infrared eclipse observations (MacQueen, 1968) point to the possibility of dust
rings forming around the Sun (Mukai and Yamamoto, 1979). Solar eclipse observations over the years suggest that
the average dust properties in the inner heliosphere change over time scales of years (Kimura et al., 1997; Ohgaito
et al., 2002). Recent whitelight observations from STEREO A (Stenborg and Howard 2017a, 2017b) provide the
shape of the F-corona and inner Zodiacal light from 5 to 24 degree line-of-sight elongation and show its flattening
to larger elongation. Closer analysis also showed that the flattening varied with spacecraft position indicating an
influence of the dust brightness near the spacecraft (Stenborg, et al.2018, Stauffer et al. 2018). Cosmic dust
particles interact with the surrounding plasma through electric charge collection, the photoelectric effect (Mann et
al. 2014), and destruction processes (sputtering, fragmentation, sublimation) and in general, near 1 AU, those
interactions little affect solar wind measurable parameters (Mann et al., 2010). This is different near the Sun. Dust
particles sublimate at bulk temperatures $\approx$ 1000– 2000 K inside $\approx$ 10 solar radii (Mukai and Mukai, 1973; Mann et
al., 2004; Mann and Murad, 2005). A fraction of dust material vaporizes during collision (Mann and Czechowski,
2005). The dust destruction rates due to sputtering are variable and increase e.g. during coronal mass ejections
(Ragot and Kahler, 2003). Solar wind particles change charge state by interaction with the dust surface or passing
through the particles (Mann et al., 2010; Minato et al., 2004). Some authors suggest interactions of newly formed
charged dust in the solar wind (Connors et al., 2014; Lai et al., 2013, 2015). Photoionisation, electron-impact





ionisation, and charge exchange quickly ionize the atoms and molecules in the solar wind (Mann and Czechowski, 2005).

From ten years of STEREO A observations attributed to nanodust impacts several important properties can be obtained. The signal explained as nanodust is 10-100 times more frequent during Stream Interaction Region (SIR)

or Interplanetary Coronal Mass Ejections (ICMEs). The observed signals exhibited a periodicity due to the crossing by STEREO of the solar magnetic equator. A correlation with solar wind perturbations, and periodicities corresponding to those of Mercury and Venus were also detected (Le Chat et al. 2015). These signals nearly disappeared on STEREO A around 2012 (Le Chat et al. 2013, Malaspina et al. 2015) when the heliosphere entered a defocusing configuration in which the nanodust coming from the inner heliosphere are pushed away from the

solar magnetic equator, therefore possibly preventing their observation. Interplanetary nanodust flux can also be measured with the RPWS instrument on board Cassini between 1 and 5 AU. Such measurements have produced two further important properties of interplanetary nanodust. Firstly, the average nanodust flux measured at 1 AU was similar in order of magnitude to the average of the highly variable flux measured by STEREO when the heliosphere was in a focusing configuration (Schippers et al., 2014), and decreased roughly as the inverse squared

heliocentric distance (Schippers et al. 2015). Secondly, the nanodust properties were shown to follow the variation in solar wind drift speed closely (Meyer-Vernet et al. 2017) as predicted by nanodust dynamics (Mann and Czechowski 2012).

The calculated trajectories for dust particles that are released in circular orbit show, except for cases when the

particles are trapped, that the particles are ejected outward. Figure 10 shows the velocity of particles that are ejected from 0.2 AU. While the particles gradually gain speed, it is apparent that in the inner solar system they still have a velocity close to that of the parent object. The same is the case for a dust particle that is released from an objected in highly elongated orbit (Fig. 11). The orbital eccentricity and perihelion assumed in this case correspond to the Aquarids meteoroids.

**7 Discussion of Implications for Observations with Parker Probe and Solar Orbiter**

The design of the radio and plasma waves instrument (RPW) on the ESA Solar Orbiter mission (Mueller and al., 2019) is similar to STEREO/WAVES. The electric Antenna system (ANT) on RPW consists of a set of three identical antennas deployed from +Z axis and from the opposite corners of the spacecraft and can operate in dipole





and monopole modes. RPW antennas consist each of a 1 metre rigid deployable boom and a 6.5 meters stacer

deployable monopole, which has a 1.5 cm radius. The Time Domain Sampler (TDS) subsystem of the RPW

instrument (Maksimovic et al., 2019) is designed to capture electromagnetic waveform snapshots at high cadence

from 200 Hz to 200 kHz, resolving in particular voltage spikes associated with interplanetary dust impacts.

Solar Orbiter will make observations of the Sun and in-situ measurements from elliptic orbits coming as close as

~60 solar radii (~0.285 AU) to the Sun. The aphelia lie outside 0.8 AU for large part of the 7 years nominal mission

time during which orbital latitude reaches 25 degree. The long cruise phase of Solar Orbiter and the elongated

spacecraft orbits with aphelia close to 1 AU provide the opportunity to study in detail the dust flux near 1AU and

to estimate the flux of sub-µm dust onto Earth, its time variation and variation during part of a solar cycle.

The FIELDS instrument on Parker Solar Probe (Bale et al. 2016) combines magnetic and electric field

measurements into a single, coordinated experiment. Four electric field antennas (2 m long, 3.18 mm diameter

Niobium C-103 thin-walled tubes) are mounted at the base of the heat shield, and deploy in full sunlight out of the

spacecraft wake, whereas a fifth antenna is mounted on the magnetometer boom in the umbra of the spacecraft.

The sensor electric field signals are transferred to a Digital Fields Board (DFB), a Time Domain Sampler (TDS)

and a Radio Frequency Spectrometer (RFS) for signal processing and digitization. The DFB and TDS make rapid

samples of waveforms with a highest sampling rate of 150,000 samples per second (DFB) and 2,000,000 samples

per second (TDS), with an on-board selection of events to reduce bit-rate. The low frequency (LF) part of the RFS

is a dual channel digital spectrometer receiving inputs from the four first antennas, either in dipole or monopole

mode, with a frequency range of 10 kHz to 2.4 MHz, allowing a relative frequency spacing of about 4.5 %.

Parker Solar Probe orbit the Sun in the ecliptic plane, making seven Venus gravity assist manoeuvres during the

seven-year nominal mission duration, which will lessen its perihelia to less than 10 $R_S$, the closest any spacecraft

has come to the Sun. In this way, the spacecraft will spend a total of 937 hours inside 20 $R_S$, 440 hours inside 15

$R_S$, and 14 hours inside 10 $R_S$ (Fox et al. 2015). The surrounding plasma changes considerably along the spacecraft

orbits (Bale et al., 2016). The orbital trajectory for the first orbit around the Sun is shown in Fig. 12.


Our considerations suggest that both RPW and FIELDS measurements in monopole mode will be able to detect

signals generated by dust impacts. Distinction between dust and other wave features needs to be considered based

on the observational data. At present, we do not know the mass range of dust particles that will be detectable. As





heliocentric distance decreases, the pulse's decay time will decrease faster than its rise time (Meyer-Vernet et al.

2017), eventually becoming smaller than the rise time, which will decrease the dust signal for large grains. The spacecraft charging and charged particles dynamics close to the Sun are expected to be considerably complicated by presence of a potential barrier (sheath structure) due to strong photoemission (Ergun et al. 2010, Campanell M. D., 2013) as well as by presence of the thermal shield and non-conducting solar panels.

Vaverka et al, (2017b) simulated pulses generated by dust impacts using a simple numerical model. The spacecraft potential is calculated using orbital-motion-limited theory and the current generated by the dust impact is represented by Gaussian function. The rise time of the pulse is estimated according to Meyer-Vernet et al. (2017) for variable photoelectron sheath. This model does not describe the detailed structure of the pulses including "pre-spikes" but only their general shapes, and can be appropriate to estimate the conditions for dust impacts.


Figure 13 shows estimated signals for impacts of 0.1 micrometre particles with speeds of 100 km/s for spacecraft at different distances from the Sun. The top panel represents temporal evolution of the spacecraft potential and bottom panel shows changes in the equilibrium potential. The charge production of the impact is assumed Q = 30 pC. With the same estimate, we find that for particle size of 0.01 micrometre and speed 200 km/s the pulses are

about a factor of 100 to 1000 smaller. It is possible to see that the amplitude and duration of the pulses are reduced with decreasing distance from the sun. This fact means that the sensitivity of dust impact detection is smaller close to the sun. It is necessary to mention that the conditions for the orbital-motion-limited theory are not satisfied near the sun. The presence of the potential barrier created due to strong photoemission, described by Ergun et al. 2010 and Campanell (2013), strongly influence the spacecraft charging and charge dynamics. An interesting result is

that the shape of the detected signal depends only weakly on the solar UV illumination which leads to the photocurrent as shown in Fig. 14 for 1 AU from the sun. The increase or decrease of the solar activity which varies the UV flux, influence the spacecraft potential but not so much the profile of the pulse generated by dust impacts.

A challenge in the data analysis will be to distinguish dust impacts signals from other events. A comparison to dust

measurements from other spacecraft should be considered. The Mercury Dust Monitor (MDM, Nogami et al. 2010) will in near future study the dust environment near Mercury at 0.31 to 0.47 AU from the Sun. Though also for those measurements, noise events are considered an issue.

*Author contribution*. This paper was prepared during discussions of the ISSI team on dust impacts at the International space
science institute in Bern Switzerland. All authors contributed to the writing of this text in an open editing process and to the
discussions that lead to the writing.

*Competing interests*. The authors declare that they have no conflict of interest.

*Data Availability*. The data will be published for open access on the UiT Open Research Repository.

*Acknowledgements*. This work is supported by the International Space Science Institute, ISSI in Bern in a team on
Dust Impacts in spacecraft. The contributions of L. N., J. V., and J. P. were supported by Czech ministry of
education youth and sport under contract LTAUSA 17066. The contributions of IM, ÅF, TA at UiT were supported
by the Research Council of Norway (grant number 262941). The contribution of SY was supported by NASA
through contract 1415150 with the Jet Propulsion Laboratory and CDAP grant 14397000. We thank Louis
Calvinhac from University Toulouse III for preparing Fig. 12 during a student project carried out at UiT. This work
was developed during several stays of some of the authors at ISSI in Bern and we wish to thank the ISSI staff for
their hospitality.

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






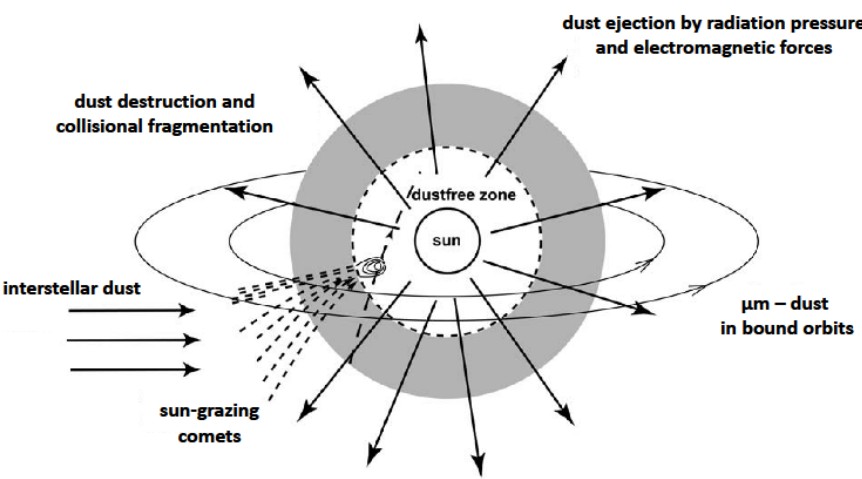

**Figure 1: Sketch of different dust components and dust interactions in the vicinity from the Sun as given in an overview (adapted from Mann et al., 2014). Recent results are presented in section 6.**

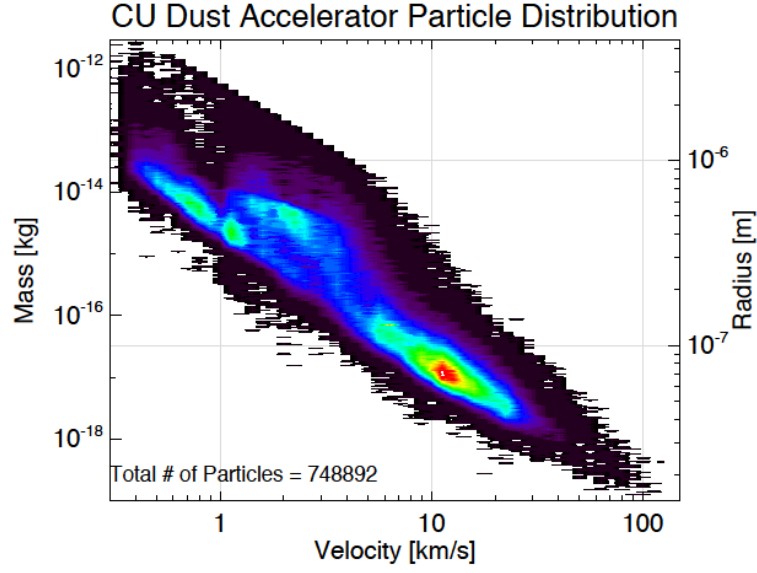


**Figure 2: The mass vs. velocity distribution of iron dust particles generated by the accelerator facility at the University of Colorado.**





**Figure 3:** This figure sketches the impact process for a spacecraft that is slightly negatively charged (left panel), zero biased (middle panel), and slightly positively charged (right panel). It is further described in the text and parameters given in Table 1.



## Dipole signals produced due to impact on dipole antenna $E_u$

A) $U_{--}$

D) $U_+$

B) $U_-$

DD) $U_+$

C) $U_0$

E) $U_{++}$

**Figure 4: Laboratory simulation of dust impacts on Cassini model showing the impact signal detected by the antenna ($E_U$ boom was bombarded) for different polarity and size of bias voltage. Different phases of dust impact signal are distinguished by colours (green – cloud generation, blue – electron escape, red – ion escape, orange – relaxation). The inserts show details of the pre-spikes (modified from Nouzák et al., 2018). The conditions in these laboratory measurements are comparable to a measurement in monopole configuration on the spacecraft.**





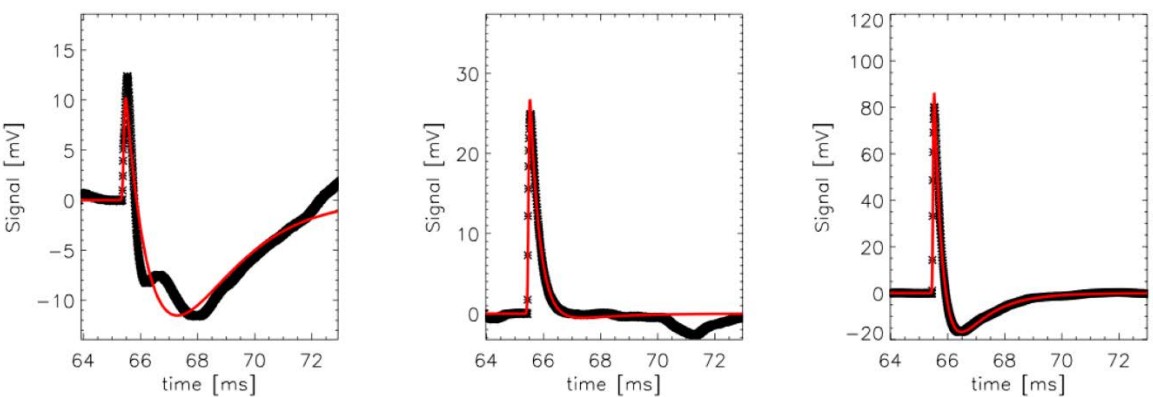

**Figure 5: Dust impact signals recorded by the STEREO/WAVES TDS on STEREO A shown with black crosses show in comparison to fit with semi-empirical model shown with red solid lines (from Zaslavsky et al. 2015).**

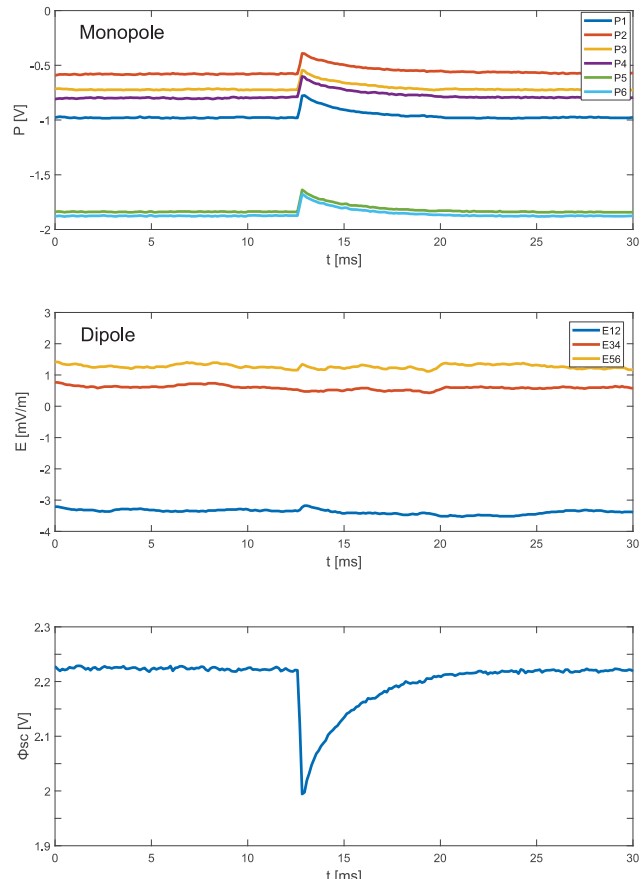


**Figure 6: A potential signal of dust impact measured with MMS. Example of a typical event related to the change of the spacecraft potential observed in monopole configuration (upper panel), the dipole measurements (middle panel) with a lack of signal and the derived voltage pulse in the lower panel (cf. Vaverka et al. 2018).**




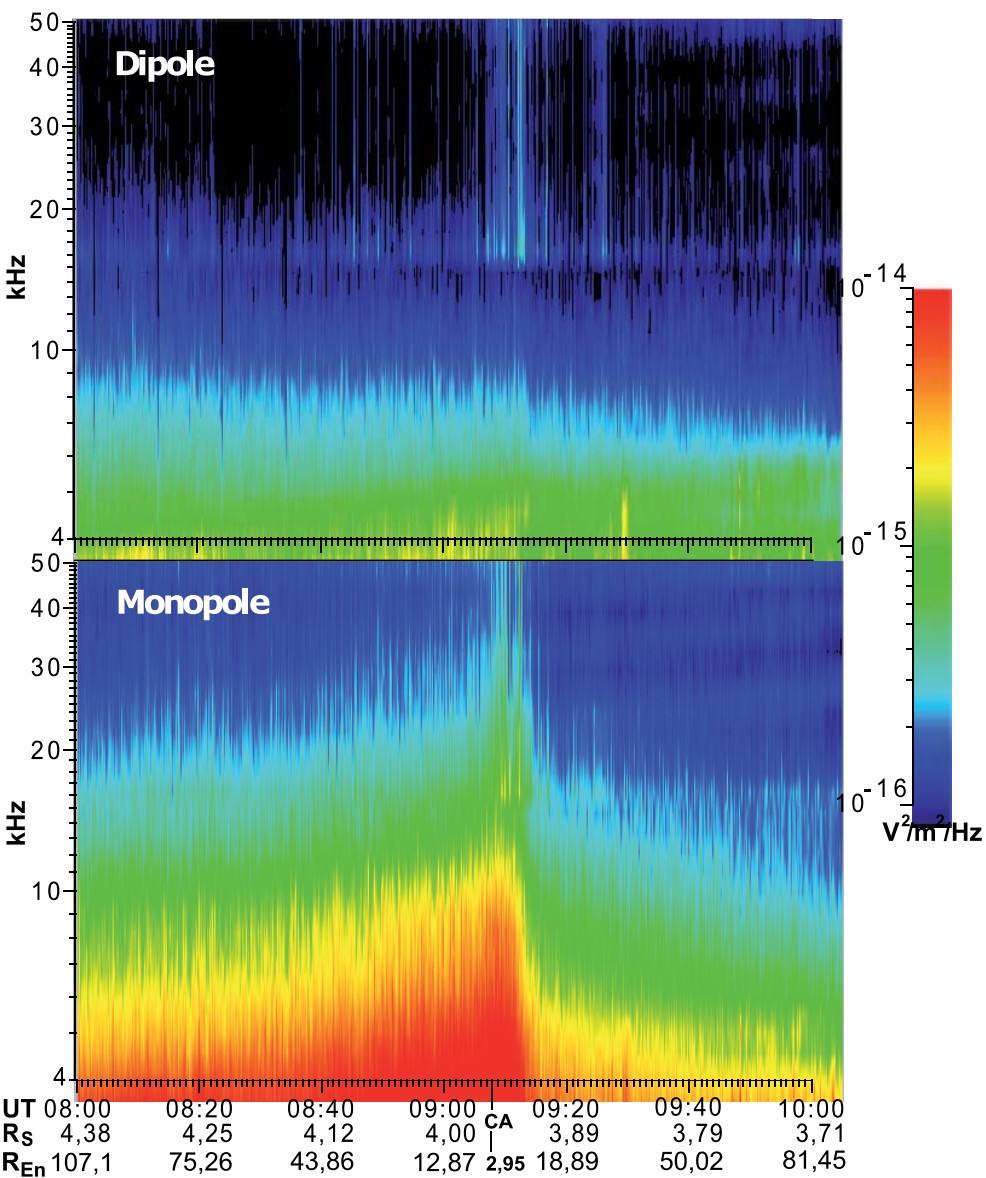

**Figure 7: From Meyer-Vernet et al. (2014): Time-frequency electric power spectral density measured by Cassini/ RPWS on 9 March 2005 in Saturn's E ring, in dipole (top) and monopole mode (bottom). The increase due to micron-sized dust impacts on the spacecraft only appears in monopole mode, whereas the dipole only measures the weaker plasma quasi-thermal and impact noise.**


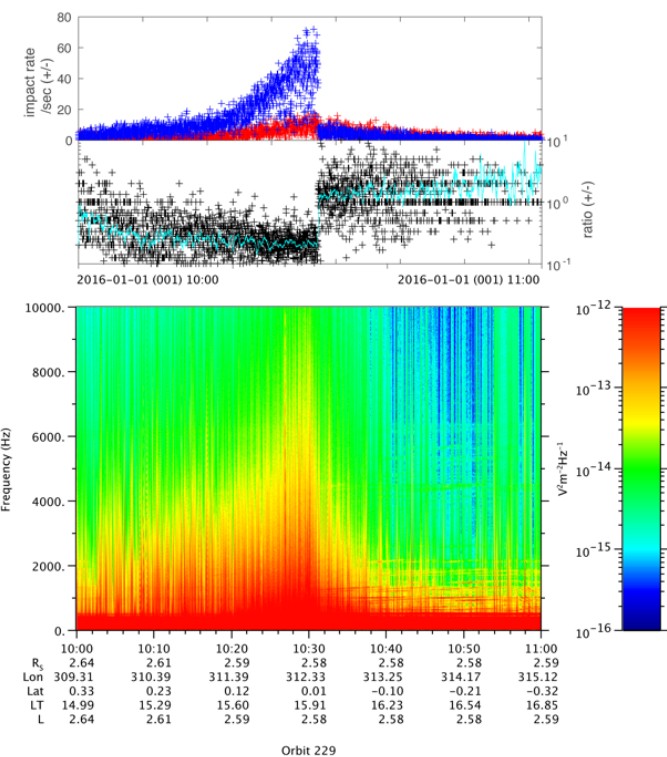

**Figure 8: Adapted from Fig. 4 and 5 of Ye et al. 2016b. RPWS wave power spectrogram around a ring plane crossing on DOY 001, 2016. The top panel shows the positive (red) and negative (blue) impact rates. The middle panel shows the impact signal polarity ratios with the moving averages (teal). At ~10:30, the antenna used was switched from monopole to dipole, which was accompanied by a decrease in the spectral power and the polarity ratio jumping back to 1.**

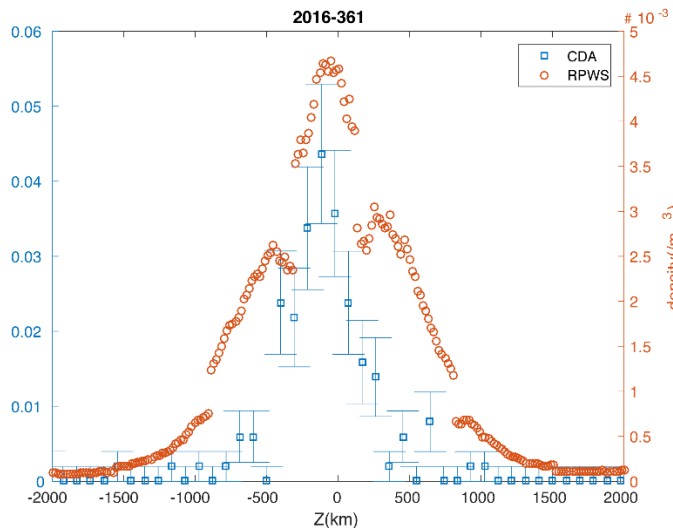

**Figure 9: Adapted from Fig. 4 of Ye et al. 2018a. Comparison of vertical dust density profiles of the Janus-Epimetheus ring measured by RPWS and CDA during the ring plane crossing on DOY 361, 2016. There is one order of magnitude difference between the two results, which is within the uncertainty limit estimated for the RPWS measurement (Ye et al. 2014).**



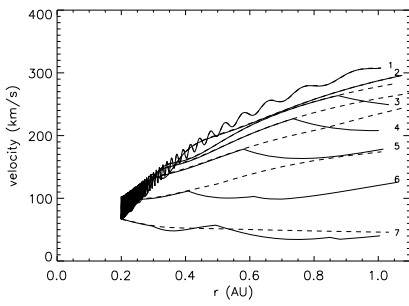


**Figure 10: Velocity as function of distance from the Sun for particles with Q/m = $10^{-4}$, $10^{-5}$, $10^{-6}$, and $10^{-7}$ e/$m_p$ released from a circular orbit with the radius 0.2 AU near the ecliptic. Solid lines correspond to the focusing, and dashed to defocusing, magnetic field orientation (adapted from Mann et al., 2014).**

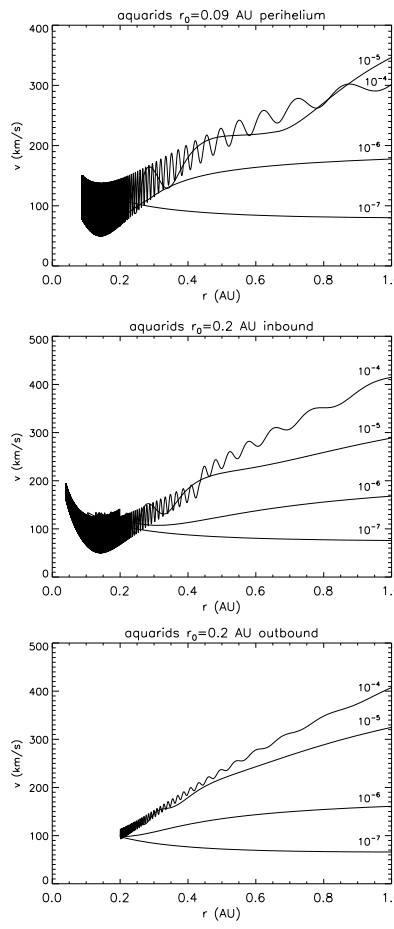

**Figure 11: Velocities of small fragments released from Aquarid meteoroids as function of the heliocentric distance. The velocities are calculated for dust with Q/m = $10^{-4}$, $10^{-5}$, $10^{-6}$ and $10^{-7}$ e/$m_p$. The ratio of surface charge to mass corresponds to sizes 3 nm, 10 nm, and larger. The particles are released from the orbit of Aquarids at the perihelion (0.09 AU from the Sun, the upper panel), and at the distance 0.2 AU on the inbound (middle panel) and outbound (lower panel) parts of the orbit. From (Czechowski and Mann, 2018)**



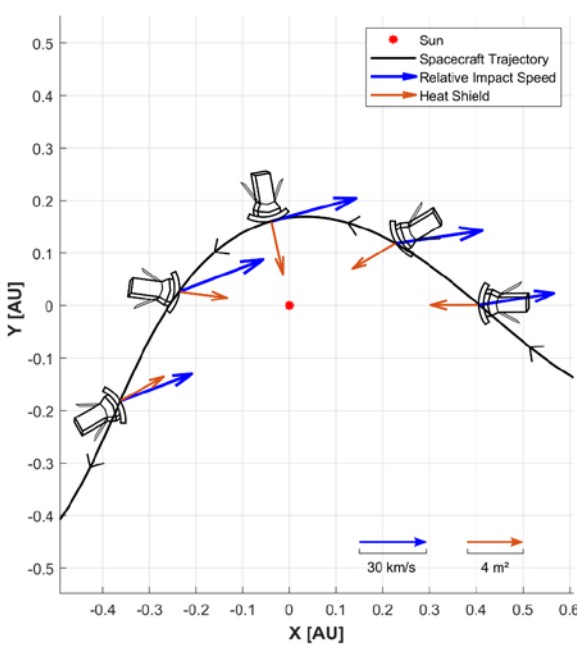

**Figure 12: Parker Solar Probe trajectory during first perihelion passage. Blue arrows indicate speed of spacecraft relative to the dust particles in circular orbit. Red arrows indicate surface vector of the spacecraft heatshield (Figure courtesy of L. Calvinhac).**

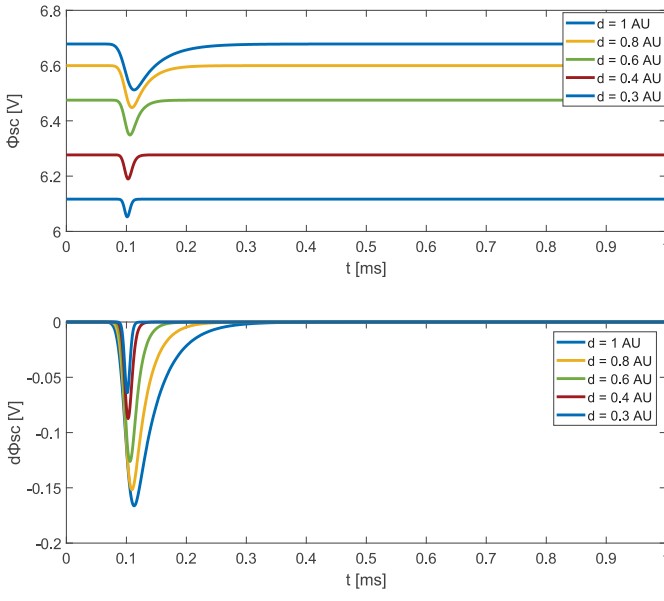

**Figure 13: The estimated signals for impacts of 0.1 micrometre particles with speed 100 km/s for spacecraft at different distance from the Sun. The charge production of the impact is assumed Q = 30 pC. Estimate based on Vaverka et al. (2017b) model.**

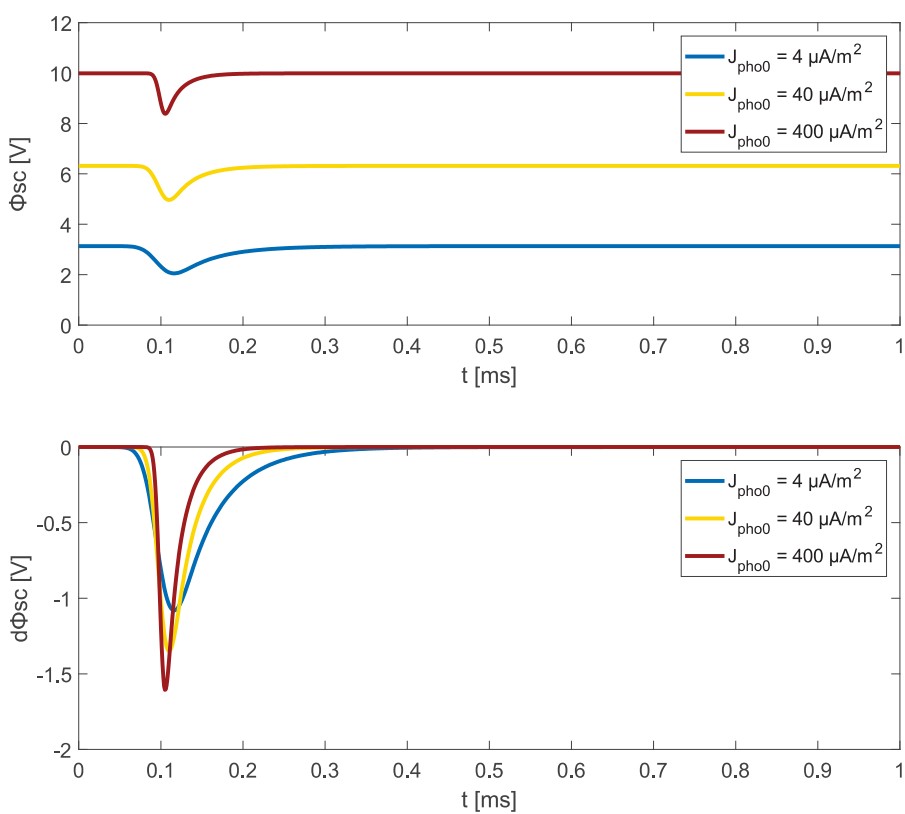

**Figure 14: The spacecraft potential change during dust impact for different values of the photocurrent. A typical value of photocurrent at 1 AU is 40 μA/m². While its influence on spacecraft potential is considerable, this is not for the change induced by impacts.**





**Table 1: Explanation of impact process signal shape illustrated in Fig. 3.**

| | **T1) cloud gen.** | **T2) electron escape** | **T3) ion escape** | **T4) relaxation** |
|---|---|---|---|---|
| **Negative spacecraft potential** ($U_{sc,0} < 0$) | Cloud generation and expansion ($U_{sc,0} < 0$) | Electron escape ($U_{sc,T2} > U_{sc,0} < 0$) | Ion escape suppressed ($U_{sc,T3} < U_{sc,T2}$) | Relaxation ($U_{sc,T4} => U_{sc,0}$) |
| **Spacecraft potential zero** ($U_{sc,0} \sim 0$) | Cloud generation and expansion ($U_{sc,0} = 0$) | Partial electron escape ($U_{sc,T2} > U_{sc,0} = 0$) | Partial ion escape ($U_{sc,T3} < U_{sc,T2}$) | Relaxation ($U_{sc,T4} => U_{sc,0}$) |
| **Positive spacecraft potential** ($U_{sc,0} > 0$) | Cloud generation and expansion ($U_{sc,0} > 0$) | Electron escape suppressed ($U_{sc,T2} > U_{sc,0} > 0$) | Ion escape ($U_{sc,T3} < U_{sc,T2}$) | Relaxation ($U_{sc,T4} => U_{sc,0}$) |

**Table 2: Examples of signal development for different bias (spacecraft potential) shown in Fig. 4.**

| | **A)** | **B)** | **C)** | **D) and DD)** | **E)** |
|---|---|---|---|---|---|
| **Process and current (positive is toward spacecraft)** | U negative $U_{--}$ | U slightly negative $U_{-}$ | U zero $U_0$ | U slightly positive $U_{+}$ | U positive $U_{++}$ |
| **Electron recollection** | zero | suppressed | partial | enhanced | full |
| **Ion recollection** | full | enhanced | partial | suppressed | zero |
| **Electron escape** | full — fast | enhanced | partial — fast | suppressed | zero |
| **Ion escape** | close to zero | suppressed | partial — slow | enhanced | full — slow |
| **Relaxation** | full | reduced | close to zero | reduced | full |
