# Peer review of "Dust observations with antenna measurements and its prospects for observations with Parker Solar Probe and Solar Orbiter"

_Annales Geophysicae, 2019_

## Referee Comment (RC1) · Anonymous Referee #1 · 13 Aug 2019

Review Comment on manuscript angeo-2019-94 entitled 'Dust observations with antenna measurements and its prospects for observations with Parker Solar Probe and Solar Orbiter' by Mann et al.,

This paper surveys the spacecraft charging observations as the dust impact detector in the past and future projects. Some minor collections and additional information for the readers will be needed before publication.

L50. 'The majority of fragments smaller than micrometers are pushed outward by radiation pressure and electromagnetic forces.' Is this the theoretical expectation or is there any observed pieces of evidence?

L 87. Fig (Fig. laboratory) -> Figure 2? Also, some text styles for figures are not set in the same format. Please check.

Chapter 5 (previous observations) This chapter summarizes the dust detection by the electric field antenna onboard different spacecraft and environment. While the author well surveys all the past dust observations, it would be helpful for future readers if the results were summarized according to the favorable electric field measurement for the dust detection of the different sizes.

L218. 'STEREO is ... orbit the Sun at around 1 AU' The unit of the orbital distances used in this chapter is sometimes AU and RâŁŹ for another time. I would suggest using only AU for easier comparison.

L208. 'Panel (E) shows a case of even higher positive bias. All electrons are re-collected in this case. The signal has no electron (blue) part and it has no "pre-spikes" in this case.' The color of the Figure 4E must be mistaken (blue->red?)

L276. 'Once these effects were taken into consideration, the estimated near-Mars micron dust flux observed by MAVEN was found to be consistent with the interplanetary dust flux expected at Mars.' Please cite the reference that shows this fact.

L315. 'Figure 9 compares ...' Please make the figure numbers in order in the text. Figure 9 cames before Figures 7 and 8.

L480. 'Figure 13 shows estimated signals for impacts of 0.1-micrometer particles with speeds of 100 km/s for spacecraft at different distances from the Sun.' Please cite the reference for figures 13 and 14.

L484. 'With the same estimate, we find that for the particle size of 0.01 micrometer and speed 200 km/s the pulses are 485 about a factor of 100 to 1000 smaller.' Is this signal difference due to the different sizes of dust? Does the signal become bigger for the higher speed with the same dust size? It would be helpful to put the result of this simulation as the bottom panel of Figure 13.

L495. 'The Mercury Dust Monitor (MDM, Nogami et al. 2010) will in near future study the dust environment near Mercury at 0.31 to 0.47 AU from the Sun.' Bepi-Colombo/MMO has also electric field antennas (PWI).
* * *

---

## Referee Comment (RC2) · Laila Andersson (Referee) · 16 Aug 2019

Main Comments: The paper collects much of the active literature into a review and the paper should be published but there are some details in the paper that needs to be straighten out first.

With out reading the paper the last two sentences in the abstract sounds contradictory and it is not clear what information is tried to be conveyed. For instance, dust impact momentum and spacecraft potential both effects the amplitude of the signal, hence it is unclear how to read the sentence. Secondly, the amount of charge production, is it as material is ejected/vaporized due to a higher EUV flux or that the dust impacts

are different. And the last sentence do not seems to provide any information for the abstract, remove last sentence from the abstract.

The paper do not discuss the 'pre-spike'. With presenting Figure 4 it needs more discussion and potential a discussion about time scales so readers can easily understand the differences. And the paper also need to discuss the timescales of the signal with respect to AC vs DC data processing. Note all laboratory analysis referred to is AC processed.

When the paper discuss how to interpret the dust signal is need to be slightly cleaner on how the develop actually is occurring. For instance, as the ejecta/vapouration occurs, is the material leaving the spacecraft neutral and as the gas expanding it become charged resulting in currents between the cloud and the spacecraft. Or is the ejecta/vapouration already charged as it leaves the spacecraft surface resulting in currents between spacecraft and infinity.

The title of the paper, the paper started as a great review of electric field measure dust. But needed with saying and SPP and SO might see dust, period. The title is '. . ..and its prospects for observations with. . .' The paper comes across with zero information of where, when and how the dust is expected to be located. When the data is analyzed if we see dust everywhere or see nothing how should that be used. The paper should either be a review article or it should provide information for future analysis. The title and abstract claims both are provided but only the first can be found in the body.

Detail comments on the text:

Page 2 line 37: typo around 'In addition'

Page 3 line 70: The first sentence has very limited with information in it. The second sentence talks about the cloud which is fine, but the first sentence is lacking of how the cloud is created.

Page 3 lie 87, typo the figure reference is wrong.

Page 4 line 107-109: This sentence depends how the authors view when the particles where ionized. If the particles are ionized directly at the surface and have momentum to leave the surface then the sentence should be rephrase. If the charge particles are created at a distance from the surface then the charge of the spacecraft might not be important because the particle already have a momentum.

Page 5, line 136: '...in the volume of initially neutral vapouration.' This needs to be tied back to the authors description of the first microseconds (initially 'neutral vapour' has not yet been mention in the paper).

Page 5 line 139: 'The first part...' is the pre-spike in many of the data. Make sure that it is clear that existing sentence is referring to the steps in Figure 3.

Page 7 line204: one sentence discuss 'pre-spike' while the next sentence say '..change the polarity ...' The 'pre-spike' do not change sight. Incorrectly stated.

Page 8 line 208: the spacecraft potential has -to first order - no effect on the pre-spike. Remove the work 'pre-spike' from this bullet.

Page 10 line 286: please discuss with the co-authors about why Wind that is normally measure as a double probe actually is an mono-pole explaining why the dust signatures are so easily to be observed.

Page 14 lines around 397-405: This section is clearly interesting the reviewer got lost in the sentences and did not get out the intended information. Please rephrase this section so it is more clearer to get the intended information out.

Page 14 line 404: The inference of sublimation of particles. There has been no discussion of and if all dust is made of what material. The discussion of sublimation needs tot be removed or need a separate paragraph so the reader can follow the authors understanding on the topic.

Page 15 line 413: the paper is about electric field instrument dust detection. Here it is stated the size of particle STEREO is observing. This needs to be explained why the

authors is claiming this. Any discussion of size and origin of dust detected by electric field instrument in this paper needs to be clearly stated of how that information was gained (other instrument, assumption, signal amplitude ignoring spacecraft potential etc).

Page 15 line 425: '. . .properties. . .follows the variation in . . .speed..' What is the properties that is discussed, shape of the dust grains?

Page 15 line 428: Bad start of the paragraph. The reviewer has no understanding of that this paragraph is saying. Remove or rewrite.

Page 17 line 475-479: More discussion of the instrument electronics respond time and coupling to the plasma is needed here. The observe rise time is effected by the electronics.

Page 17 line 494 – 497: the paper is about electric field measurement, why has the MDM instrument been called out and not other dedicate dust detectors? Remove the paragraph, distracting the reader from the topic.

---

## Author Comment (AC1) · 1 Oct 2019

We thank the referee for the constructive comments. All comments were taken into account and the manuscript was modified accordingly. A response to each comment is given below in bold letters.

Review Comment on manuscript angeo-2019-94 entitled 'Dust observations with antenna measurements and its prospects for observations with Parker Solar Probe and Solar Orbiter' by Mann et al., This paper surveys the spacecraft charging observations as the dust impact detector in the past and future projects. Some minor collections and additional information for the readers will be needed before publication.

[Figure]

L50. 'The majority of fragments smaller than micrometers are pushed outward by radiation pressure and electromagnetic forces.' Is this the theoretical expectation or is there any observed pieces of evidence? -This is based on theoretical expectations and to some extent confirmed by observations. We have included a reference describing the theory and obesevations.

L 87. Fig (Fig. laboratory) -> Figure 2? Also, some text styles for figures are not set in the same format. Please check. -We corrected the reference to the figure and modified Fig.2.

Chapter 5 (previous observations) This chapter summarizes the dust detection by the electric field antenna onboard different spacecraft and environment. While the author well surveys all the past dust observations, it would be helpful for future readers if the results were summarized according to the favorable electric field measurement for the dust detection of the different sizes. -We modified the structure of the chapter 5 and introduced a new order of subchapters in order to have a better flow of the description.

L218. 'STEREO is . . . orbit the Sun at around 1 AU' The unit of the orbital distances used in this chapter is sometimes AU and RâŁZ' for another time. I would suggest using only AU for easier comparison. -AU describes a distance from the Sun and Re a distance from the Earth. We used AU for spacecraft in interplanetary space (STEREO, Cassini cruise) and Earth radii (Re) for Earth-orbiting spacecraft (Cluster) and Wind located at L1 point. Difference between AU and Re is more than three orders of magnitude and we therefore think that using these two units in different sections of the text is appropriate and helps the reader to quickly understand the size scales.

L208. 'Panel (E) shows a case of even higher positive bias. All electrons are re-collected in this case. The signal has no electron (blue) part and it has no "pre-spikes" in this case.' The color of the Figure 4E must be mistaken (blue->red?) -We have changed the panel E in Figure 4.

L276. 'Once these effects were taken into consideration, the estimated near-Mars

micron dust flux observed by MAVEN was found to be consistent with the interplanetary dust flux expected at Mars.' Please cite the reference that shows this fact. -We added a reference: indeed the experiment analysis indeed supports that the measured impacts are from interplanetary dust.

L315. 'Figure 9 compares . . .' Please make the figure numbers in order in the text. Figure 9 cames before Figures 7 and 8. -We checked the order of figures and modified it according to sequence given in the text.

L480. 'Figure 13 shows estimated signals for impacts of 0.1-micrometer particles with speeds of 100 km/s for spacecraft at different distances from the Sun.' Please cite the reference for figures 13 and 14. -These figures are originally created for this manuscript. We have modified the text so that this fact is obvious for the readers.

L484. 'With the same estimate, we find that for the particle size of 0.01 micrometer and speed 200 km/s the pulses are 485 about a factor of 100 to 1000 smaller.' Is this signal difference due to the different sizes of dust? Does the signal become bigger for the higher speed with the same dust size? It would be helpful to put the result of this simulation as the bottom panel of Figure 13. -We have modified the sentence. We describe general dependence on mass and velocity of impacting grains instead of one particular case.

L495. 'The Mercury Dust Monitor (MDM, Nogami et al. 2010) will in near future study the dust environment near Mercury at 0.31 to 0.47 AU from the Sun.' Bepi-Colombo/MMO has also electric field antennas (PWI). -We included information about the electric field antenna (PWI) into the text.

---

## Author Comment (AC2) · 1 Oct 2019

We thank the referee for the constructive comments. All comments were taken into account and the manuscript was modified accordingly. A response to each comment is given below in bold letters.

Main Comments: The paper collects much of the active literature into a review and the paper should be published but there are some details in the paper that needs to be straighten out first. With out reading the paper the last two sentences in the abstract sounds contradictory and it is not clear what information is tried to be conveyed. For instance, dust impact momentum and spacecraft potential both effects the amplitude

[Figure]

of the signal, hence it is unclear how to read the sentence. Secondly, the amount of charge production, is it as material is ejected/vaporized due to a higher EUV flux or that the dust impacts are different. And the last sentence do not seems to provide any information for the abstract, remove last sentence from the abstract.

->We removed the last sentence of the abstract and modified the one but last sentence.

The paper do not discuss the 'pre-spike'. With presenting Figure 4 it needs more discussion and potential a discussion about time scales so readers can easily understand the differences. And the paper also need to discuss the timescales of the signal with respect to AC vs DC data processing. Note all laboratory analysis referred to is AC processed.

->"Pre-spikes" are explained and discussed in sections 3 and 4. We add information about the time scales to section 3 and about influence of the instrument electronics to section 4.

When the paper discuss how to interpret the dust signal is need to be slightly cleaner on how the develop actually is occurring. For instance, as the ejecta/vapouration occurs, is the material leaving the spacecraft neutral and as the gas expanding it become charged resulting in currents between the cloud and the spacecraft. Or is the ejecta/vapouration already charged as it leaves the spacecraft surface resulting in currents between spacecraft and infinity.

->The charge production is a result of the impact process and in fact, some recombination occurs in the initial stage after the impact. To better describe the process we have revised the first part of section 2 and also included new references.

The title of the paper, the paper started as a great review of electric field measure dust. But needed with saying and SPP and SO might see dust, period. The title is '. . ..and its prospects for observations with. . .' The paper comes across with zero information of where, when and how the dust is expected to be located. When the data is analyzed

if we see dust everywhere or see nothing how should that be used.

->The paper should either be a review article or it should provide information for future analysis. The title and abstract claims both are provided but only the first can be found in the body. The paper covers the dust observations with antenna measurements, as well as the prospects for observations with Parker Probe and Solar Orbiter. The latter are addressed in section 6 and section 7.

Detail comments on the text: Page 2 line 37: typo around 'In addition'.

->We have corrected the text.

Page 3 line 70: The first sentence has very limited with information in it. The second sentence talks about the cloud which is fine, but the first sentence is lacking of how the cloud is created.

->We have revised this paragraph and also changed the sequence of the content, so that now we mention in the second sentence in more detail how the impact vapour cloud is generated.

Page 3 line 87, typo the figure reference is wrong.

->We have corrected the reference.

Page 4 line 107-109: This sentence depends how the authors view when the particles where ionized. If the particles are ionized directly at the surface and have momentum to leave the surface then the sentence should be rephrase. If the charge particles are created at a distance from the surface then the charge of the spacecraft might not be important because the particle already have a momentum. The heading of this subsection was changed to "Impact signal generation" so that it better describes the content of this subsection. The detail description of plasma cloud generation is described earlier in the text in section "Dust impact ionisation".

Page 5, line 136: '. . .in the volume of initially neutral vapouration.' This needs to be

tied back to the authors description of the first microseconds (initially 'neutral vapour' has not yet been mention in the paper). We have rephrased this part and we now describe the volume ionization in section 2.

Page 5 line 139: 'The first part. . .' is the pre-spike in many of the data. Make sure that it is clear that existing sentence is referring to the steps in Figure 3.

->Reference on Fig. 3 was included to the sentence to make sure that the sentence refers to steps in Fig. 3.

Page 7 line204: one sentence discuss 'pre-spike' while the next sentence say '..change the polarity . . .' The 'pre-spike' do not change sign. Incorrectly stated.

->We agree with the referee statement that the 'pre-spike' do not change sign. However, the signal polarity can change sign according to different balance between escaped electrons and ions. We have replaced word "pre-spike" by "signal".

Page 8 line 208: the spacecraft potential has -to first order - no effect on the pre-spike. Remove the work 'pre-spike' from this bullet.

->The laboratory data in Fig. 4 clearly show significant effect of spacecraft potential on pre-spike. The pre-spike part of the signal decreases with increasing size of positive spacecraft potential. This effect is also illustrated in the sketch of Fig. 3.

Page 10 line 286: please discuss with the co-authors about why Wind that is normally measure as a double probe actually is an mono-pole explaining why the dust signatures are so easily to be observed.

->Explanation of Wind data as a monopole was misunderstanding. Wind dipole antenna (different than dipole probes as on MMS or Cluster) can detect dust impact on the spacecraft body by asymmetrical influence of the spacecraft potential and expanding plasma cloud on both parts of the dipole.

Page 14 lines around 397-405: This section is clearly interesting the reviewer got lost

in the sentences and did not get out the intended information. Please rephrase this section so it is more clearer to get the intended information out.

->We have re-written this section.

Page 14 line 404: The inference of sublimation of particles. There has been no discussion of and if all dust is made of what material. The discussion of sublimation needs tot be removed or need a separate paragraph so the reader can follow the authors understanding on the topic. We have re-written this section. The information on dust sublimation can be found in an earlier review that we cite in the text.

Page 15 line 413: the paper is about electric field instrument dust detection. Here it is stated the size of particle STEREO is observing. This needs to be explained why the authors is claiming this. Any discussion of size and origin of dust detected by electric field instrument in this paper needs to be clearly stated of how that information was gained (other instrument, assumption, signal amplitude ignoring spacecraft potential etc).

->We have re-written this entire section and discuss STEREO observations to illustrate complexity of nanodust trajectories.

Page 15 line 425: '. . .properties. . .follows the variation in . . .speed..' What is the properties that is discussed, shape of the dust grains?

->We have re-written this entire section and this was revised.

Page 15 line 428: Bad start of the paragraph. The reviewer has no understanding of that this paragraph is saying. Remove or rewrite.

->We have re-written this entire section and this was revised.

Page 17 line 475-479: More discussion of the instrument electronics respond time and coupling to the plasma is needed here. The observe rise time is effected by the electronics.

->We agree, the observed rise time of the signal is affected by the electronics of antenna instrument and it is also often limited by it. We modified the text and added a discussion about signal detection, response time and coupling in section 3.

Page 17 line 494 – 497: the paper is about electric field measurement, why has the MDM instrument been called out and not other dedicate dust detectors? Remove the paragraph, distracting the reader from the topic.

->We have modified the text. The meaning of this paragraph is that the comparison with another instruments detecting dust should be considerate. MDM and PWI instruments on board BepiColombo will be able to detect dust impacts at similar distance from the Sun as Parker Solar Probe and Solar Orbiter. This is reason why we mentioned this particular mission. We also now point out the importance of overlapping observations to measure the same dust flux with different instruments from different spacecraft